# Enhanced C/EBPβ function promotes hyperplastic versus hypertrophic fat tissue growth and prevents steatosis in response to high-fat diet feeding

Christine Müller[1†], Laura M Zidek[2†], Sabrina Eichwald[2], Gertrud Kortman[1], Mirjam H Koster[3], Cornelis F Calkhoven[1,2]*

[1]European Research Institute for the Biology of Ageing, University Medical Center Groningen, University of Groningen, Groningen, Netherlands; [2]Leibniz Institute on Aging - Fritz Lipmann Institute, Jena, Germany; [3]Division Molecular Genetics, Department of Pediatrics, University Medical Center Groningen, University of Groningen, Groningen, Netherlands

**Abstract** Chronic obesity is correlated with severe metabolic and cardiovascular diseases as well as with an increased risk for developing cancers. Obesity is usually characterized by fat accumulation in enlarged – hypertrophic – adipocytes that are a source of inflammatory mediators, which promote the development and progression of metabolic disorders. Yet, in certain healthy obese individuals, fat is stored in metabolically more favorable hyperplastic fat tissue that contains an increased number of smaller adipocytes that are less inflamed. In a previous study, we demonstrated that loss of the inhibitory protein-isoform C/EBPβ-LIP and the resulting augmented function of the transactivating isoform C/EBPβ-LAP promotes fat metabolism under normal feeding conditions and expands health- and lifespan in mice. Here, we show that in mice on a high-fat diet, LIP-deficiency results in adipocyte hyperplasia associated with reduced inflammation and metabolic improvements. Furthermore, fat storage in subcutaneous depots is significantly enhanced specifically in LIP-deficient male mice. Our data identify C/EBPβ as a regulator of adipocyte fate in response to increased fat intake, which has major implications for metabolic health and aging.

*For correspondence:
c.f.calkhoven@umcg.nl

†These authors contributed equally to this work

## Editor's evaluation

This study provides important insight into the mechanisms involved in regulating the response to an obesity-inducing diet in mice. This study demonstrates that C/EBPβ acts as a key protective factor against many of the negative consequences of a high-fat diet in mice and further clarifies the downstream processes involved. Obesity is an already significant and growing health problem, and this work may help identify new strategies to combat obesity going forward.

## Introduction

Nutrient overload, particularly in combination with a sedentary lifestyle, is the main cause of the increasing incidence of obesity we are facing today. In most cases, chronic obesity provokes the development of metabolic diseases like insulin resistance, type 2 diabetes (T2D), non-alcoholic fatty liver disease (NFALD), and non-alcoholic steatohepatitis (NASH) (*Longo et al., 2019*). The surplus of fat is stored in the white adipose tissue (WAT) largely without an increase in adipocyte number, resulting in an increase in fat cell size. This hypertrophy is accompanied with reduced vascularization and oxygen

supply, and an increase in macrophage infiltration, and inflammation (*Frasca et al., 2017*; *Tchkonia et al., 2010*). Since the storage capacity of the hypertrophic cells is limited, fat starts to accumulate in ectopic tissues like liver, heart and skeletal muscle (*Frasca et al., 2017*). This steatosis, also referred to as 'lipotoxicity' further promotes metabolic disorders. However, there is an exception from this scenario as individuals exist that are chronically obese but stay – at least transiently – metabolically healthy. Evidence from mouse and a few human studies suggest that storing surplus of nutrients as fat through adipocyte hyperplasia (increasing number) is associated with metabolic health (*White and Ravussin, 2019*). As the fat storage can be distributed over more fat cells, the individual fat cells stay smaller, are metabolically more active, and less inflamed (*Ghaben and Scherer, 2019*). So far not much is known about the underlying molecular mechanisms and involved regulators that drive fat storage in either the hypertrophic or the hyperplastic direction. Such regulators may be attractive targets to therapeutically switch the adipocytes from a hypertrophic into a hyperplasic state in order to prevent metabolic complications associated with obesity.

CCAAT/Enhancer Binding Protein beta (C/EBPβ) is a transcription factor known to regulate adipocyte differentiation together with other C/EBPs and peroxisome proliferator-activated receptor gamma (PPARγ) (*Siersbæk and Mandrup, 2011*). In all cases of C/EBPβ controlled cellular processes, it is important to consider that different protein isoforms of C/EBPβ exist. The two long C/EBPβ isoforms, LAP1 and LAP2 (Liver-enriched activating proteins) differ slightly in length and both function as transcriptional activators. The N-terminally truncated isoform LIP (Liver-enriched inhibitory protein) acts inhibitory because it lacks transactivation domains yet binds to DNA in competition with LAP1/2 (*Descombes and Schibler, 1991*). We have shown earlier that LIP expression is stimulated by mTORC1 signaling involving a *cis*-regulatory short upstream open reading frame (uORF) in the Cebpb-mRNA (*Calkhoven et al., 2000*; *Zidek et al., 2015*).

Mutation of the uORF in mice (*Cebpb^{ΔuORF}* mice) results in loss of LIP expression, unleashing LAP transactivation function, resulting in C/EBPβ super-function. (*Müller et al., 2018*; *Wethmar et al., 2010*; *Zidek et al., 2015*) In *Cebpb^{ΔuORF}* mice, metabolic and physical health is preserved and maintained during ageing with features also observed under calorie restriction (CR), including leanness, enhanced fatty acid oxidation, prevention of steatosis, better insulin sensitivity and glucose tolerance, preservation of motor coordination, delayed immunological ageing and reduced interindividual variation in gene expression of particularly metabolic genes (*Müller et al., 2018*; *Zidek et al., 2015*).

Here, we show that C/EBPβ is critically involved in dictating the adipocyte phenotype and the metabolic outcome in response to high-fat diet (HFD) feeding in mice. C/EBPβ super-function in *Cebpb^{ΔuORF}* mice causes fat to accumulate in hyperplastic rather than hypertrophic depots. In addition, *Cebpb^{ΔuORF}* male mice store the surplus of fat more efficiently in subcutaneous fat stores. Accordingly, the *Cebpb^{ΔuORF}* mice are protected against the development of steatosis and better maintain glucose tolerance, indicating a healthy obese phenotype.

## Results

Separate cohorts of male and female *Cebpb^{ΔuORF}* mice and wild-type (wt) control littermates on a C57BL/6 J background were fed a high-fat diet (HFD; 60% fat) or standard chow (10% fat) for a period of 19 weeks. In males, both genotypes gained weight over the whole experimental period yet with significantly lower body weights for *Cebpb^{ΔuORF}* mice, suggesting that they are partially protected from HFD induced weight gain (*Figure 1A*). Both the food intake and the energy efficiency (energy that was extracted from the food during digestion) were similar in *Cebpb^{ΔuORF}* and wt males (*Figure 1— figure supplement 1A, B*). Lean mass and fat mass body composition was determined by computer tomography (CT) after 19 weeks of HFD feeding. On standard chow, fat mass of *Cebpb^{ΔuORF}* males was reduced compared to wt males (*Figure 1B*) as we showed before (*Zidek et al., 2015*). Unexpectedly, we observed a slight but significantly increased overall fat mass volume in the *Cebpb^{ΔuORF}* males compared to wt controls (*Figure 1B*), which correlates with a relative reduction in lean mass (*Figure 1—figure supplement 1C*). There was no significant difference in the amount of visceral fat mass between wt and *Cebpb^{ΔuORF}* males under HFD (*Figure 1C*). However, the HFD fed *Cebpb^{ΔuORF}* males had a significantly increased subcutaneous fat mass compared to the wt controls (*Figure 1D*). Taken together, the data show that *Cebpb^{ΔuORF}* male mice store more fat upon HFD feeding compared to wt littermates, and that this extra fat is mainly stored in the subcutaneous fat depot. Similar to male mice, the body weight gain of *Cebpb^{ΔuORF}* females after 19 weeks on HFD was smaller compared to

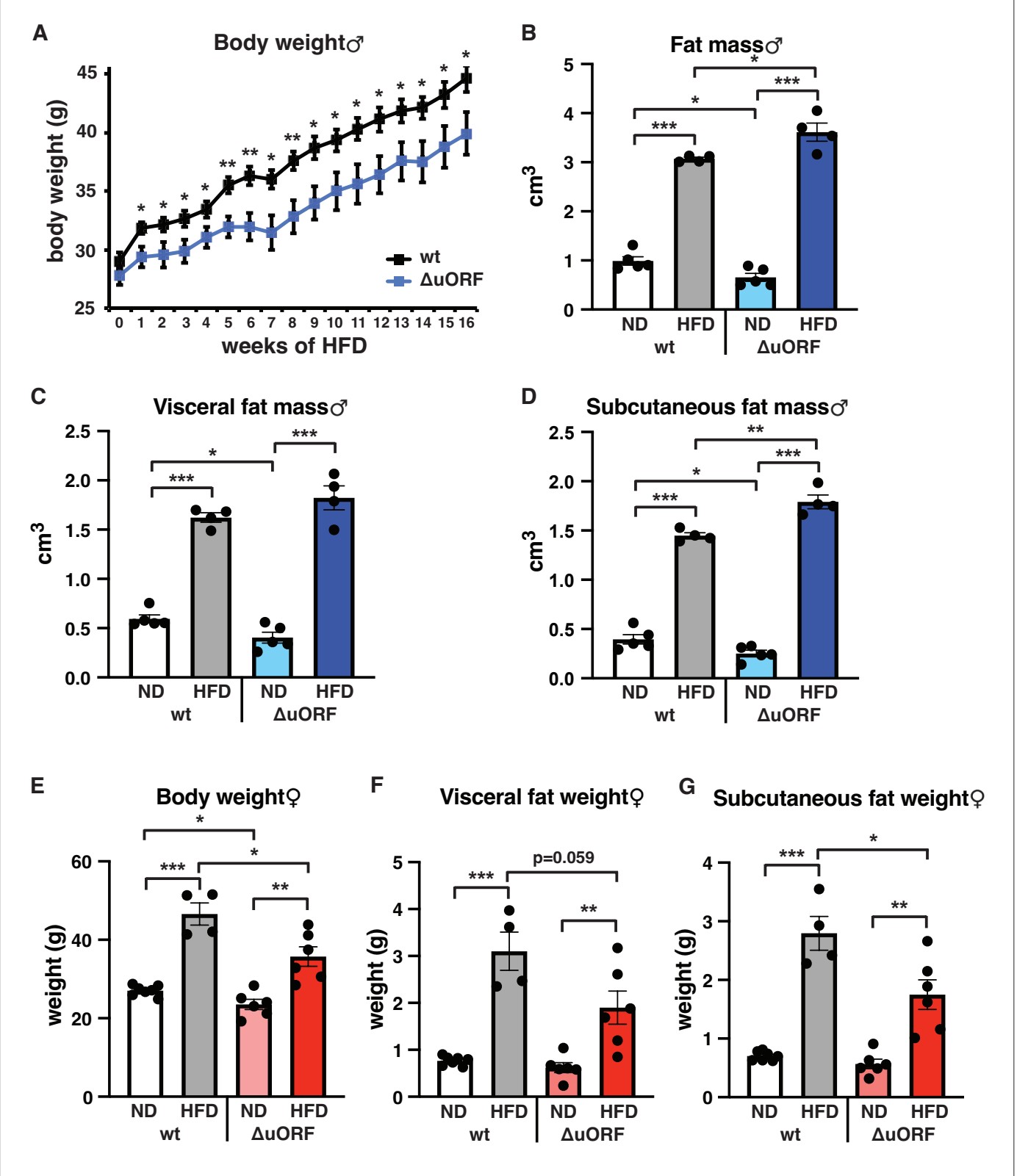

**Figure 1.** *Cebpb^ΔuORF* mice on high-fat diet (HFD). (**A**) Growth curves of wt and *Cebpb^ΔuORF* (ΔuORF) male mice on HFD (wt, n = 10; *Cebpb^ΔuORF*, n = 8). (**B**) Volume of total fat mass as measured by abdominal CT analyses (males,19 weeks; ND, n = 5; HFD, n = 4). (**C**) Volume of visceral fat mass as measured by abdominal CT analyses (males,19 weeks; ND, n = 5; HFD, n = 4) (**D**) Volume of subcutaneous fat mass as measured by abdominal CT analyses (males, 19 weeks; ND, n = 5; HFD, n = 4). (**E**) Female body weight (week 19; ND wt, n = 7; HFD wt, n = 4; ND and HFD ΔuORF, n = 6). (**F**) Visceral fat weight

*Figure 1 continued on next page*

Figure 1 continued

(females, week 19; ND wt, n = 7; HFD wt, n = 4; ND and HFD ΔuORF, n = 6). (**G**) Subcutaneous fat weight (females, week 19; ND wt, n = 7; HFD wt, n = 4; ND and HFD ΔuORF, n = 6). All values are mean ± SEM. p-Values were determined with Student's t-test, *p < 0.05; **p < 0.01; ***p < 0.001.

The online version of this article includes the following source data and figure supplement(s) for figure 1:

**Source data 1.** Raw data related to *Figure 1A–G*.

**Figure supplement 1.** Food intake, energy efficiency and lean mass of male mice on high-fat diet (HFD).

**Figure supplement 1—source data 1.** Raw data and calculations related to *Figure 1A–C*.

the weight gain of wt females (*Figure 1E*). However, different from the male mice, in females this was associated with a reduced accumulation of fat particularly in the subcutaneous depot in *Cebpb*^ΔuORF compared to wt females as was determined by fat tissue weight (*Figure 1F, G*).

Next, we compared histological sections from visceral fat of HFD fed *Cebpb*^ΔuORF mice and wt littermates and observed that the adipocyte size in the *Cebpb*^ΔuORF mice for both sexes is significantly smaller compared to the wt mice (*Figure 2A, B*). Calculated from visceral fat volume (CT analysis) of males or visceral fat weight of females and the average adipocytes area (histology) per mouse, the number of adipocytes is approximately 3 times higher in *Cebpb*^ΔuORF males and 1.5 times higher in *Cebpb*^ΔuORF females compared to their wt littermates.

Adipose tissue composed of small adipocytes is metabolically more active and better supplied with oxygen, and its inflammatory state is usually lower than that of enlarged adipocytes (*Ghaben and Scherer, 2019*). We therefore analyzed the inflammatory state of visceral white adipose tissue (WAT) by determining the expression of the macrophage marker CD68 and the inflammatory cytokines TNFα, MCP1, IL-1 and IL-6 using quantitative PCR (qPCR) and immunohistochemistry (anti-CD68). Male wt mice on HFD show a strong increase in CD68 mRNA expression compared to wt males on ND indicating increased macrophage infiltration. In contrast, CD68 mRNA expression is much lower in the visceral fat from HFD fed *Cebpb*^ΔuORF males and not significantly different from ND fed mice (*Figure 3A*). Accordingly, histological staining of visceral WAT derived from three different mice shows more pronounced macrophage infiltration in wt mice on HFD (19 weeks) compared to *Cebpb*^ΔuORF males (*Figure 3B*). In addition, expression of the inflammatory markers TNFα, MCP1, IL-1β, and IL-6 was significantly induced in the visceral fat of wt males on HFD (*Figure 3C*). In contrast, in *Cebpb*^ΔuORF

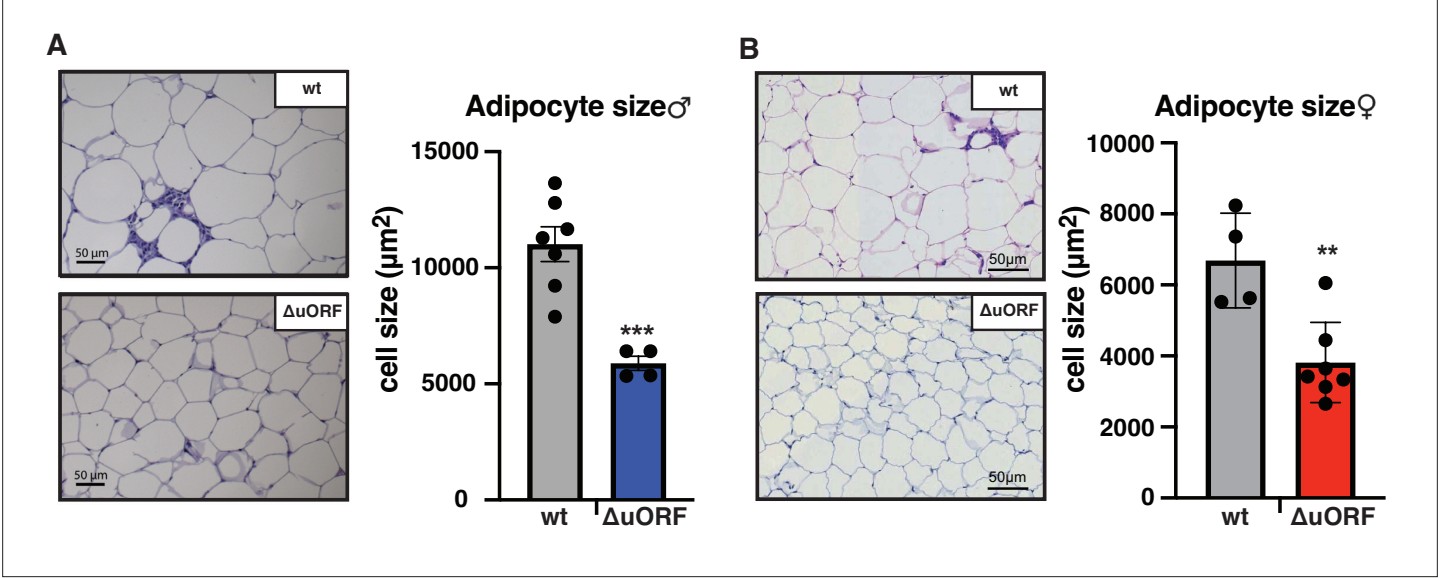

**Figure 2.** *Cebpb*^ΔuORF mice on high-fat diet (HFD) store fat in hyperplastic adipocytes. Histological hematoxylin and eosin (H&E) staining of epididymal WAT from (**A**) males (19 weeks HFD) and (**B**) females (19 weeks HFD). Quantification of the fat cell area is shown at the right (males: wt, n = 7; ΔuORF, n = 4; females: wt, n = 4; ΔuORF, n = 7; 12 adjacent cells are measured per mouse).

The online version of this article includes the following source data for figure 2:

**Source data 1.** Raw data related to *Figure 2A, B*.

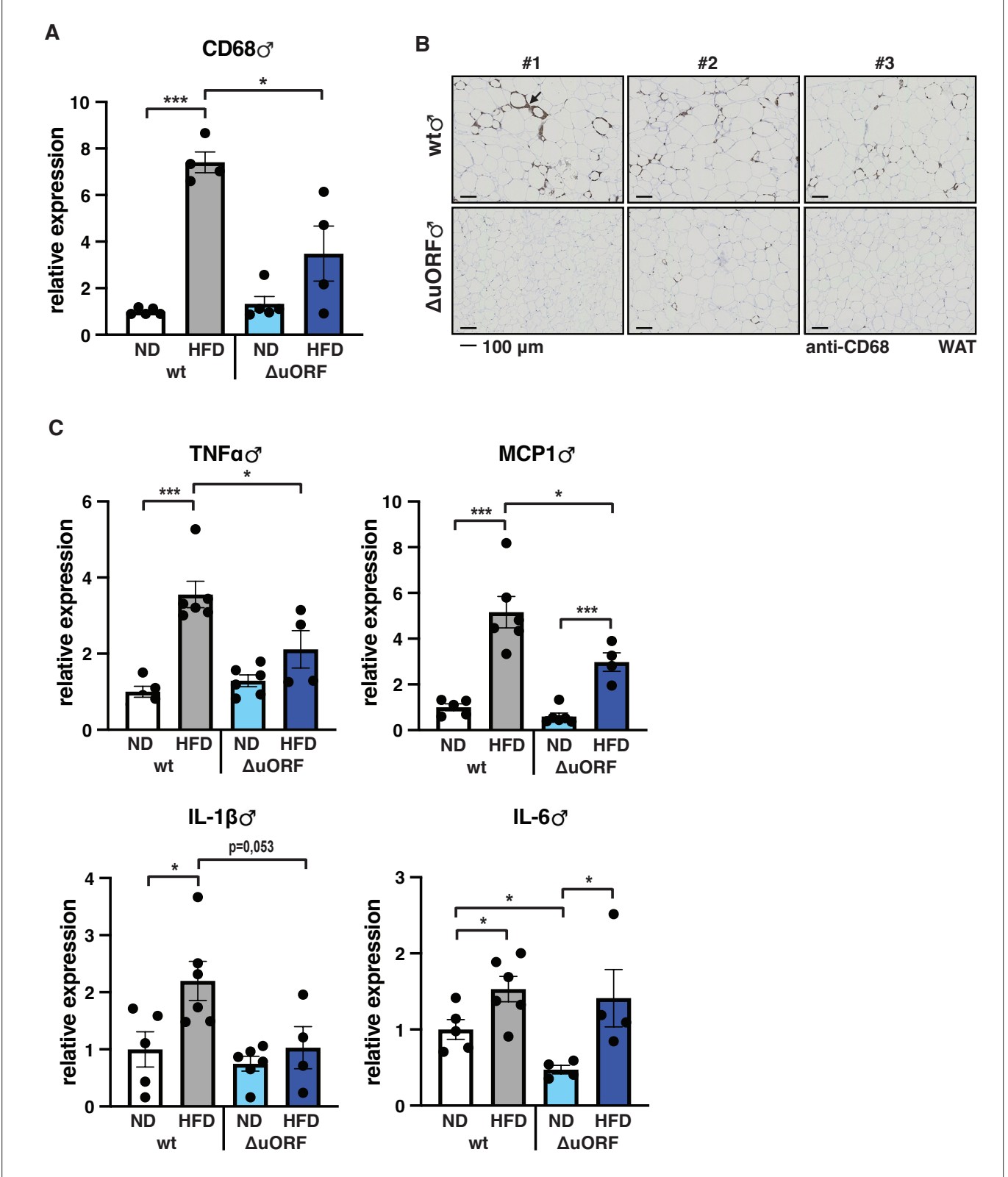

**Figure 3.** Inflammation of the visceral WAT is reduced in *Cebpb^ΔuORF* male mice on high-fat diet (HFD). (**A**) Relative mRNA expression levels of the macrophage marker CD68 measured in the visceral fat of *Cebpb^ΔuORF* (ΔuORF) and wt male mice on either normal diet (ND) or HFD (19 weeks; ND, n = 5; HFD, n = 4). (**B**) Immunohistological staining of the visceral fat of *Cebpb^ΔuORF* male mice (ΔuORF) and wt mice on HFD (19 weeks) using a CD68-specific antibody (arrow points to specific staining). Histological sections from three individual mice per genotype are shown. (**C**) Relative mRNA expression

*Figure 3 continued on next page*

Figure 3 continued

levels of the inflammatory cytokines TNFα, MCP1, IL-1β, and IL6 measured in the visceral fat of *Cebpb^ΔuORF* (ΔuORF) and wt male mice on either normal diet (ND) or HFD (19 weeks; wt: ND, n = 5; HFD, n = 6; ΔuORF: ND, n = 6 (for IL-6, n = 4, the results of two mice were excluded due to undetectable signal); HFD, n = 4). All values are mean ± SEM. p-Values were determined with Student's t-test, *p < 0.05; ***p < 0.001.

The online version of this article includes the following source data for figure 3:

**Source data 1.** Raw data related to *Figure 3A, C*.

males, HFD feeding did not significantly induce TNFα and IL-1β expression, and the induction of MCP1 was significantly lower compared to HFD fed wt males. Solely IL-6 levels were comparable between the two genotypes on HFD feeding. For female wt mice the mean value of induction of CD68 expression in response to HFD is high but strongly varies between the mice. Therefore, the lower value of CD68 expression in HFD fed *Cebpb^ΔuORF* mice is not statistically significant yet shows much less variation (*Figure 4A*). Notwithstanding, the anti-CD68 staining of visceral WAT did show infiltration by macrophages in wt females on HFD although to a lower extend compared to wt males, while in *Cebpb^ΔuORF* females hardly any staining was observed (*Figure 4B*). The inflammatory markers TNFα, MCP1 and IL-1 are all induced in wt and *Cebpb^ΔuORF* mice on HFD feeding to similar extends and thus no differences are measured between the genotypes. No induction was observed for IL6 expression in both genotypes upon HFD feeding (*Figure 4C*).

Together, these data demonstrate that *Cebpb^ΔuORF* mice on HFD feeding store extra fat through an increase in adipocyte numbers (hyperplasia), which results in smaller sized adipocytes and reduced fat tissue inflammation in males. In *Cebpb^ΔuORF* females, adipocytes are also smaller but consistent differences in the inflammation state of the visceral fat were not measured.

Adipocyte hypertrophy under obese conditions is associated with a limit in fat storage capacity of the adipocytes and enhanced lipolysis (*Khan et al., 2009*; *Laurencikiene et al., 2011*) The resulting increase in the concentration of fatty acids in the circulation causes lipid accumulation in non-fat tissues like liver, muscle and heart (*Longo et al., 2019*). In a previous study, we showed that *Cebpb^ΔuORF* males on a C57Bl/6 genetic background are protected against age-related steatosis on a ND, compared to wt mice that do accumulate fat in the liver at an age of 8 months (*Zidek et al., 2015*). To investigate possible differences in HFD-induced steatosis, we stained histological sections of liver for fat accumulation. Livers of both male and female wt mice showed massive fat accumulation (steatosis) on HFD. Compared to the wt mice, fat accumulation was much lower in *Cebpb^ΔuORF* mice of both sexes (*Figure 5A, B*). In agreement with the differences in steatosis, the livers of wt females on HFD are significantly heavier than livers of *Cebpb^ΔuORF* females while in males a trend towards heavier wt livers is visible (*Figure 5C, D*). Fat accumulation on HFD was also lower in the heart and skeletal muscle of male mice (*Figure 5—figure supplement 1A*) and for both males and females the weight of the heart on HFD is higher in wt compared to *Cebpb^ΔuORF* mice (*Figure 1—figure supplement 1B*). Altogether, these data show that *Cebpb^ΔuORF* mice are protected against steatosis in the liver and other organs in response to HFD.

Chronic obesity often results in the loss of glucose homeostasis (*Abranches et al., 2015*). We therefore analyzed glucose tolerance and insulin sensitivity in the HFD fed *Cebpb^ΔuORF* mice and wt littermates. Glucose clearance from the circulation measured by intraperitoneal glucose tolerance test (IPGTT) was impaired in response to 7 weeks HFD feeding for the wt mice of both sexes, as is shown by a significantly increased area under the curve (AUC) (*Figure 6A, B*). For the *Cebpb^ΔuORF* male mice, the already significantly better glucose clearance on normal diet does not change on HFD (*Figure 6A*). The female *Cebpb^ΔuORF* mice on HFD show reduced glucose clearance in the IPGTT compared to ND but they perform significantly better than the HFD fed wt females (*Figure 6B*). At the time of 7 weeks on HFD, both the wt and *Cebpb^ΔuORF* mice of both sexes did not develop insulin insensitivity as measured by intraperitoneal insulin sensitivity test (IPIST) (*Figure 6C, D*). Both the *Cebpb^ΔuORF* males and females, however, generally performed better on IPIST than the wt mice.

In conclusion, our data show that *Cebpb^ΔuORF* mice on HFD feeding perform better in a glucose tolerance test, are protected against steatosis and show a lower inflammatory status of WAT, although the latter is less evident in females. These metabolically favorable phenotypes of the *Cebpb^ΔuORF* mice correlate with hyperplastic fat storage and in male mice with more efficient fat accumulation in the subcutaneous depot.

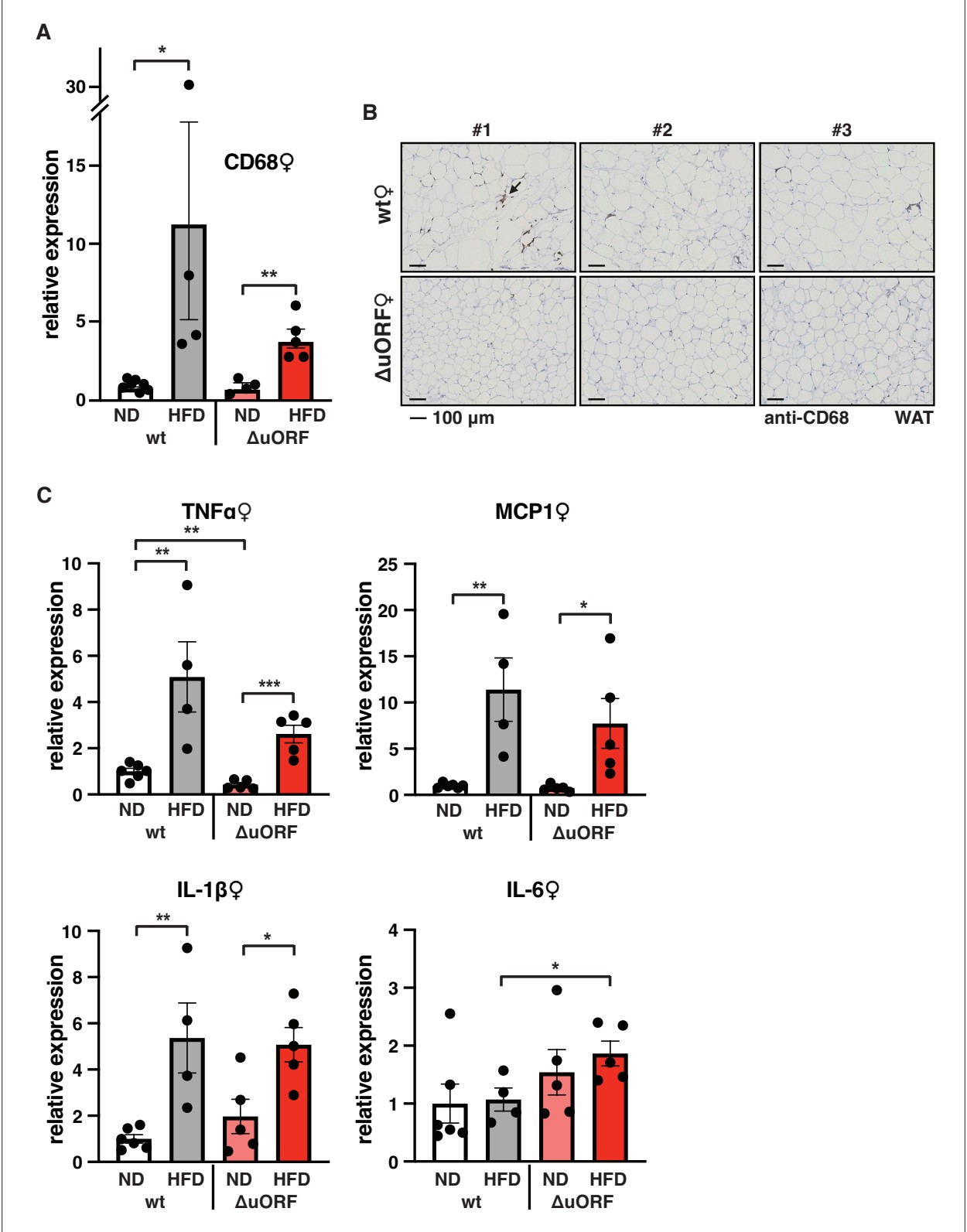

**Figure 4.** Macrophage infiltration of the visceral WAT is reduced in *Cebpb^{ΔuORF}* female mice on high-fat diet (HFD). (**A**) Relative mRNA expression levels of the macrophage marker CD68 measured in the visceral fat of *Cebpb^{ΔuORF}* (ΔuORF) and wt female mice on either normal diet (ND) or HFD (19 weeks; wt: ND, n = 7; HFD, n = 4; ΔuORF: ND, n = 4 (the result from one mouse was excluded due to undetectable signal); HFD, n = 5). (**B**) Immunohistological staining of the visceral fat of *Cebpb^{ΔuORF}* female mice (ΔuORF) and wt mice on HFD (19 weeks) using a CD68-specific antibody (arrow points to specific

*Figure 4 continued on next page*

Figure 4 continued

staining). Histological sections from three individual mice per genotype are shown. (**C**) Relative mRNA expression levels of the inflammatory cytokines TNFα, MCP1, IL-1β, and IL6 measured in the visceral fat of *Cebpb^ΔuORF* female mice (ΔuORF) and wt mice on either normal diet (ND) or HFD (19 weeks; wt: ND, n = 6; HFD, n = 4; ΔuORF: ND, n = 5; HFD, n = 5). All values are mean ± SEM. p-Values were determined with Student's t-test, *p < 0.05; **p < 0.01.

The online version of this article includes the following source data for figure 4:

**Source data 1.** Raw data related to *Figure 4A, C*.

---

C/EBPβ is a known transcriptional regulator of fat cell differentiation and function (*Siersbæk and Mandrup, 2011*). We have shown earlier that the truncated C/EBPβ isoform LIP inhibits adipocyte differentiation and that fibroblasts derived from *Cebpb^ΔuORF* mice have an increased adipogenic differentiation potential (*Zidek et al., 2015*). We therefore analyzed the expression in the visceral WAT of the adipogenic transcription factors C/EBPα and PPARγ, the sterol regulatory element-binding protein 1 c (SREBP1c) as a key transcription factor for lipogenesis, and fatty acid synthase (FAS) as a key lipogenic enzyme, by quantitative PCR. In wt male mice all four transcripts are significantly lower expressed on HFD compared to ND (*Figure 7A*). This generally corresponds to their protein levels as determined by immunoblotting, although the expression of PPARγ and SREBP1c varies considerably between the mice (*Figure 7—figure supplement 1A*). In the WAT of *Cebpb^ΔuORF* males on ND, expression of the four transcripts is similar (C/EBPα and PPARγ) or higher compared to WAT from wt males on ND (SREBP1 and FAS, also shown in *Zidek et al., 2015*), and its reduction under HFD occurs to a lesser extend compared to wt mice (*Figure 7A*). With the exception of C/EBPα, the transcript levels generally correspond to the protein levels, although here variations of in particular PPARγ and SREBP1 levels complicate interpretation (*Figure 7—figure supplement 1A*). For the wt female mice, only the transcript levels of FAS were downregulated upon HFD and for *Cebpb^ΔuORF* mice only expression of PPARγ and FAS was significantly higher on HFD compared to wt females on HFD (*Figure 7B*). The better maintained expression of FAS in HFD fed *Cebpb^ΔuORF* females in the qPCR analysis however could not be recapitulated with immunoblotting (*Figure 7—figure supplement 1B*).

To examine whether HFD feeding is associated with changes in LAP/LIP expression, we analyzed extracts from WAT isolated from wt and *Cebpb^ΔuORF* mice of both sexes under ND and HFD conditions. In wt males, both LAP and LIP isoforms were upregulated upon HFD feeding as shown in the immunoblot (*Figure 8A*) and determined by quantification of blot signals from a cohort (*Figure 8B, C*). The quantification reveals a small but significant decrease in the LAP/LIP ratio (*Figure 8B*), indicating that the inhibitory function of LIP increases upon HFD in wt males. Due to the LIP-deficiency caused by the *Cebpb^ΔuORF* mutation, the LAP/LIP ratio is very high for the *Cebpb^ΔuORF* males and does not change upon HFD feeding (some residual LIP expression is usually visible in *Cebpb^ΔuORF* mice due to leaky scanning over the not-optimal AUG-start codons for the LAP proteins). For the females, a significant increase in both LAP and LIP expression in response to HFD feeding was only observed in the *Cebpb^ΔuORF* mice (*Figure 8D, F*). However, no significant changes were measured in LAP/LIP expression ratios, despite an overall trend towards a lower LAP/LIP ratio on HFD in wt females (*Figure 8D, E*). Taken together, LAP/LIP isoform ratios decline in response to HFD feeding due to higher increase of LIP expression which is significant for males but not for females.

## Discussion

The *Cebpb^ΔuORF* mutation prevents expression of the inhibitory C/EBPβ protein isoform LIP which results in unconstrained function of the C/EBPβ transactivator isoform LAP. In two previous reports, we have shown that *Cebpb^ΔuORF* mice display metabolic improvements and a delay in the onset of age-related conditions, collectively resulting in an extended lifespan in females (*Müller et al., 2018*; *Zidek et al., 2015*). Here, we demonstrate that *Cebpb^ΔuORF* mice are protected against the development of metabolic disturbances in response to HFD feeding. In males, this improved metabolic phenotype occurs although the total fat mass in *Cebpb^ΔuORF* mice is increased in response to HFD to a greater extent than in wt mice. Our data indicate that two special features of the white adipose tissue (WAT) in *Cebpb^ΔuORF* males contribute to these metabolic improvements. Firstly, *Cebpb^ΔuORF* males on a HFD store the surplus of nutrients in fat depots that expand through hyperplasia; they increase the number of adipocytes and thus the individual cells have to store less fat. These smaller adipocytes

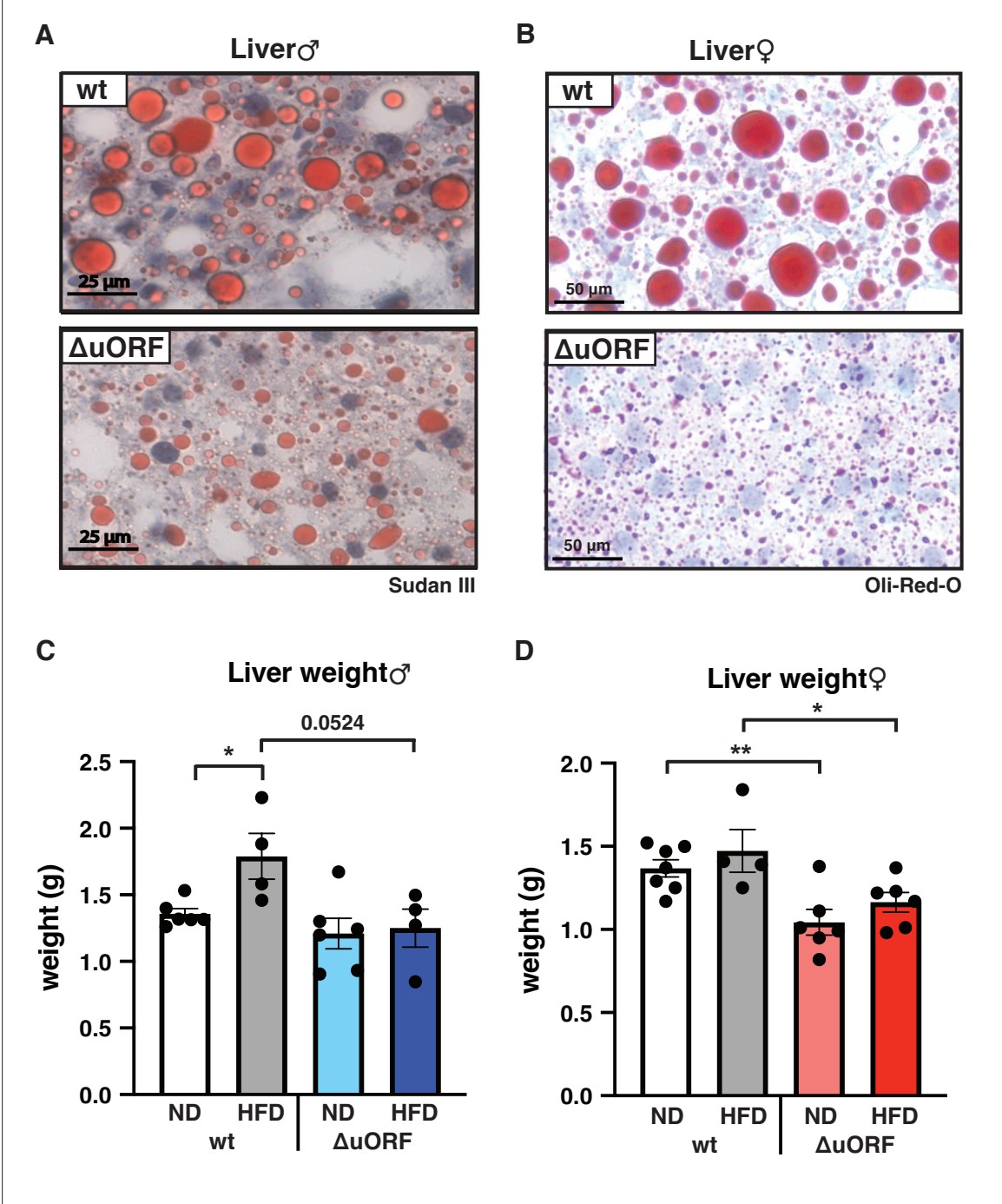

**Figure 5.** *Cebpb^{ΔuORF}* mice on high-fat diet (HFD) are protected against steatosis. Histological sections of liver from (**A**) males and (**B**) females of wt or *Cebpb^{ΔuORF}* mice (ΔuORF) (19 weeks). Sections were stained with hematoxylin (blue) and Sudan III (males) or Oil-Red-O (females) for red color lipid staining. Liver weight of (**C**) males and (**D**) females of wt or *Cebpb^{ΔuORF}* mice (ΔuORF) (19 weeks; males: ND, n = 6; HFD, n = 4; females: wt ND, n = 7, wt HFD, n = 4; ΔuORF wt and HFD, n = 6). All values are mean ± SEM. p-Values were determined with Student's t-test, *p < 0.05; **p < 0.01.

The online version of this article includes the following source data and figure supplement(s) for figure 5:

**Source data 1.** Raw data related to *Figure 5C, D*.

**Figure supplement 1.** *Cebpb^{ΔuORF}* mice on high-fat diet (HFD) are protected against steatosis in the heart and skeletal muscle.

**Figure supplement 1—source data 1.** Raw data related to *Figure 5—figure supplement 1C, D*.

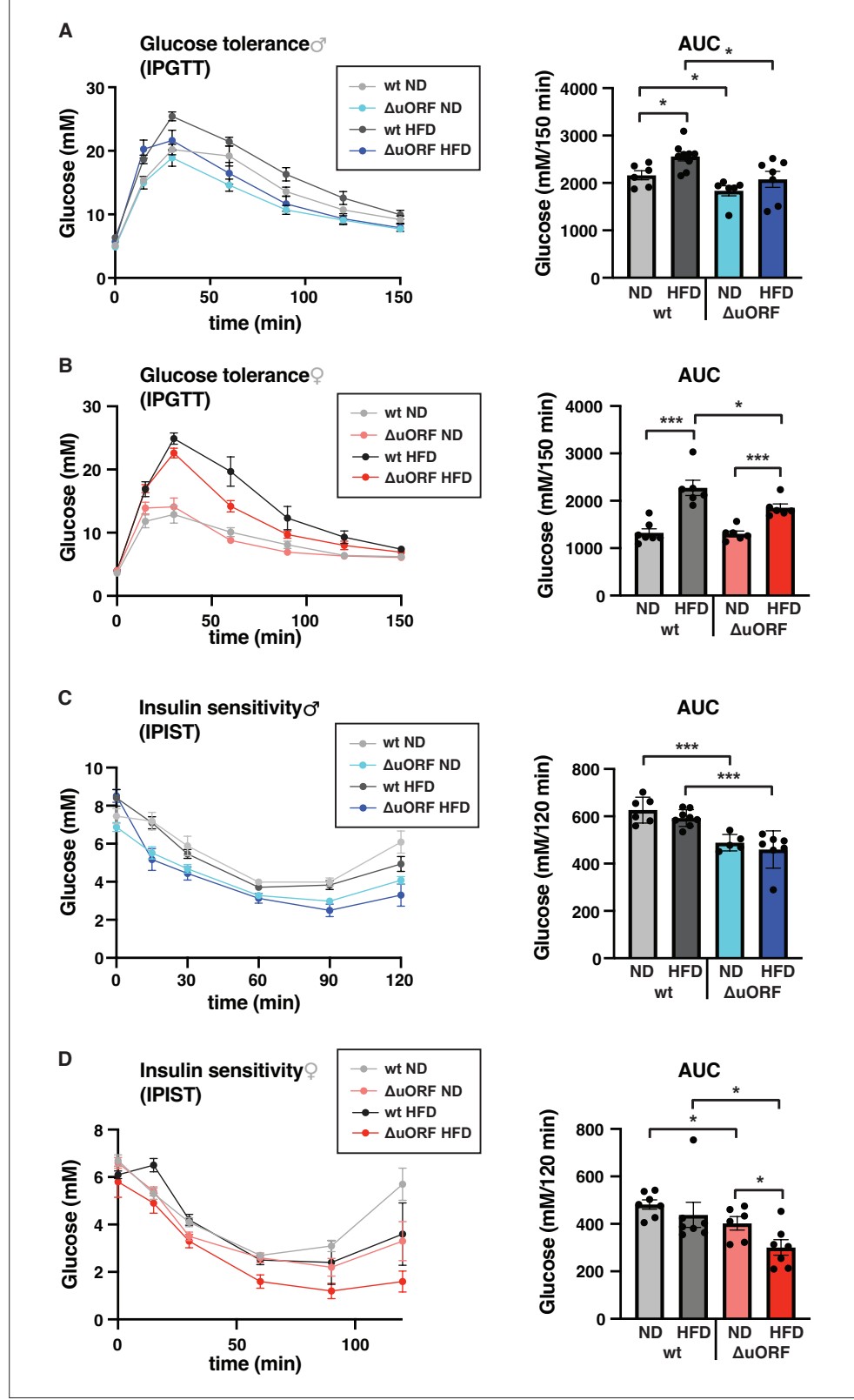

**Figure 6.** *Cebpb^{ΔuORF}* mice show improved glucose tolerance and insulin sensitivity on a high-fat diet (HFD). Intraperitoneal glucose tolerance test (IPGTT) with the calculated area under the curve (AUC) of *Cebpb^{ΔuORF}* (**A**) male and (**B**) female (ΔuORF) and wt mice injected i.p. with glucose (2 g/kg) after a 16 hr fast (7 weeks; males: wt ND, n = 6; wt HFD, n = 9; ΔuORF ND, n = 6; ΔuORF HFD, n = 7; females: wt ND, n = 7; wt HFD, n = 6; ΔuORF

*Figure 6 continued on next page*

*Figure 6 continued*

ND and HFD, n = 6). Intraperitoneal insulin sensitivity test (IPIST) with the calculated area under the curve (AUC) of *Cebpb^ΔuORF* (**C**) male and (**D**) female (ΔuORF) mice and wt mice injected i.p. with insulin (0.5 IU/kg) (7 weeks; males: wt ND, n = 6; wt HFD, n = 8; ΔuORF ND, n = 5; ΔuORF HFD, n = 7; females: wt ND, n = 7; wt HFD, n = 7; ΔuORF ND, n = 6; ΔuORF HFD, n = 7). All values are mean ± SEM. p-Values were determined with Student's t-test, *p < 0.05; ***p < 0.001.

The online version of this article includes the following source data for figure 6:

**Source data 1.** Raw data related to *Figure 6A–D*.

---

are metabolically more active and less inflamed compared to the inflated wt adipocytes residing in a hypertrophic fat depot. Hypertrophic adipocytes are known to secrete inflammatory cytokines that promote insulin resistance and other metabolic disturbances (*Reilly and Saltiel, 2017*; *Weisberg et al., 2003*). Furthermore, since the number of adipocytes in hypertrophic fat tissue does not increase and the amount of fat that can be stored in an adipocyte is limited, fat starts to accumulate in ectopic tissues like liver or muscle, compromising metabolic health (*Frasca et al., 2017*). Accordingly, in wt males on HFD we observed pronounced inflammation and macrophage infiltration in the visceral WAT and severe steatosis. In contrast, *Cebpb^ΔuORF* males on HFD are protected against these metabolic disturbances, which also correlated with better maintenance of glucose tolerance. We have shown earlier that the expression of genes related to fatty acid oxidation is enhanced in the liver of *Cebpb^ΔuORF* mice accompanied by a significant increase in fatty acid oxidation (*Zidek et al., 2015*). This enhanced fat utilization likely contributes to the reduced lipid accumulation in the liver of *Cebpb^ΔuORF* mice on HFD, and the healthier metabolic phenotype of *Cebpb^ΔuORF* mice is presumably the result of the combination of an increase in WAT function and fat utilization. In addition, the *Cebpb^ΔuORF* males store relatively more fat in the subcutaneous compartment than wt mice, which relieves the fat storage pressure for the visceral depots. Fat storage in the subcutaneous fat depot is associated with a better metabolic health status in humans and mice, while fat storage in the visceral fat depot is associated with insulin resistance and inflammation (*Carey et al., 1997*; *McLaughlin et al., 2011*; *Tran et al., 2008*). In contrast to the males, female *Cebpb^ΔuORF* mice showed reduced fat accumulation in the subcutaneous fat depot upon HFD compared to wt mice, and the visceral fat depot showed a trend towards a reduced fat storage although this difference was not statistically significant (*Figure 1F and G*). However, similar to the *Cebpb^ΔuORF* males, the adipocyte cell size in the visceral fat from *Cebpb^ΔuORF* females was significantly reduced and the calculated number of adipocytes was higher compared to wt females revealing increased adipocyte hyperplasia also in HFD fed *Cebpb^ΔuORF* females. Accordingly, also the female mice showed an improved metabolic phenotype on HFD including reduced hepatic steatosis and better maintained glucose tolerance. The difference in inflammation between wt and *Cebpb^ΔuORF* females was less pronounced compared to males and only visible in antibody staining of the macrophage marker CD68 indicating reduced macrophage infiltration in the visceral fat of *Cebpb^ΔuORF* females. However, macrophage infiltration seemed to be less pronounced in wt females on HFD than in wt males based on the CD68 immunohistological staining (compare *Figures 3B and 4B*), which might be explained by the known anti-inflammatory function of β-estradiol (*Camporez et al., 2019*). The generally lower vulnerability for inflammation in females may mitigate the differences in inflammatory cytokines between the two genotypes.

The HFD induced adipocytic hyperplasia in *Cebpb^ΔuORF* mice indicates that unconstrained LAP functionality – through loss of inhibitory function of LIP – stimulates adipocyte differentiation and function. It may explain why *Cebpb^ΔuORF* male mice store more fat in WAT on a HFD than wt littermates assuming that efficient fat storage by adaptive increase of the number of adipocytes prevents relocation of fat to peripheral tissues. Although fat storage in female *Cebpb^ΔuORF* mice upon HFD was rather reduced compared to wt littermates, also their adipocyte numbers in the visceral fat depot were increased. These observations are in line with our previous experiments showing that mouse embryonic fibroblasts (MEFs) derived from *Cebpb^ΔuORF* mice are much more efficiently induced to undergo adipogenesis than wt MEFs, and differentiation of 3T3-L1 preadipocytes is strongly suppressed upon ectopic induction of LIP (see data in Expanded View Figure 3 B, C of *Zidek et al., 2015*). Furthermore, the Kirkland lab has shown that endogenous LIP levels increase upon ageing in pre-adipocytes and in isolated fat cells from rats which correlated with reduced C/EBPα expression and reduced differentiation potential of the pre-adipocytes and that overexpression of LIP in preadipocytes from young

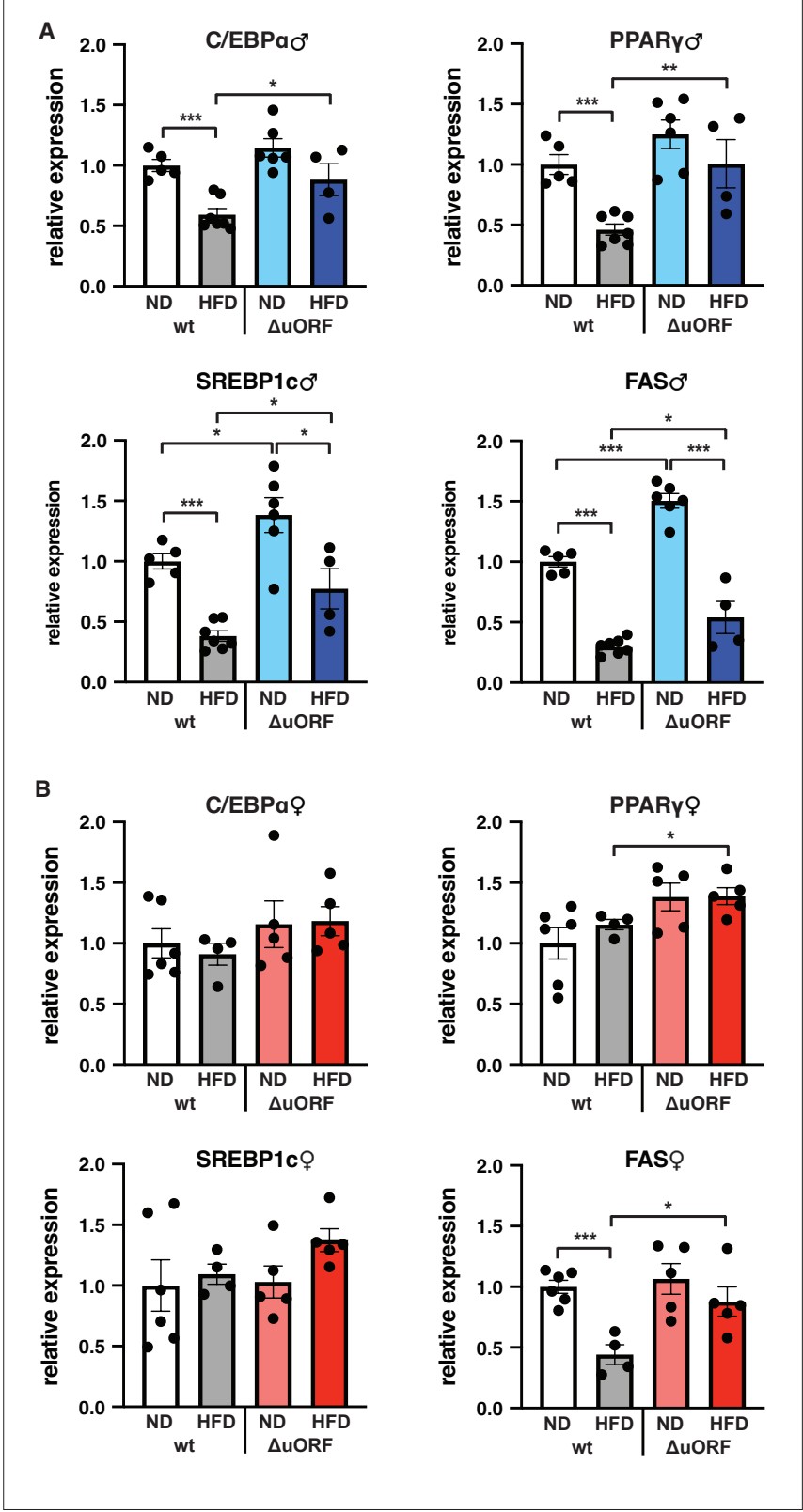

**Figure 7.** Expression of key adipogenic genes is elevated in *Cebpb^ΔuORF* male mice on high-fat diet (HFD). Relative mRNA expression levels of the adipogenic transcription factors C/EBPα, PPARγ and SREBBP1c and key enzyme FAS measured visceral WAT of *Cebpb^ΔuORF* (**A**) male and (**B**) female (ΔuORF) mice and wt mice on either normal diet (ND) or HFD (19 weeks; males: wt ND, n = 5; wt HFD, n = 7; ΔuORF ND, n = 6; ΔuORF HFD, n = 4; females: wt

*Figure 7 continued on next page*

*Figure 7 continued*

ND, n = 6; wt HFD, n = 4; ΔuORF ND and HFD, n = 5). All values are mean ± SEM. p-Values were determined with Student's t-test, *p < 0.05; **p < 0.01; ***p < 0.001.

The online version of this article includes the following source data and figure supplement(s) for figure 7:

**Source data 1.** Raw data related to *Figure 7A, B*.

**Figure supplement 1.** Protein expression of key adipogenic genes is elevated in *Cebpb^ΔuORF* mice on high-fat diet (HFD).

---

rats impaired adipogenesis (*Karagiannides et al., 2001*). Similarly, we observed a shift of the C/EBPβ isoform ratio towards more LIP expression in visceral fat from wt males on HFD (*Figure 8*) which probably inhibits adipogenesis and thereby might contribute to adipocyte hypertrophy observed in wt mice.

In the current model of the regulatory cascade of adipocyte differentiation C/EBPβ and C/EBPδ induce the expression of C/EBPα and PPARγ, which by positive feedback stimulate each other's expression (*Siersbæk and Mandrup, 2011*). Pharmacological activation of PPARγ by thiazolidines similarly to the *Cebpb^ΔuORF* mutation stimulates adipocyte differentiation, results in fat storage in hyperplastic adipocytes and in a shift to fat storage in the subcutaneous compartment, resulting in improved metabolic health (*Adams et al., 1997*; *Fujiwara et al., 1988*; *McLaughlin et al., 2010*; *Okuno et al., 1998*). Our finding that the mRNA expression of adipogenic transcription factors PPARγ, SREBP1 and C/EBPα in the visceral fat of *Cebpb^ΔuORF* males is better maintained on HFD compared to wt mice also fits to these observations. However, in contrast to our qPCR data, C/EBPα protein levels were rather downregulated in *Cebpb^ΔuORF* males both at ND and HFD conditions upon HFD (*Figure 7—figure supplement 1*) suggesting counter-regulation at the level of translation. This might be an adaptive response to increased LAP function yet does not seem to affect the adipocyte hyperplasia phenotype.

In the visceral WAT of wt females, LIP levels and the LAP/LIP isoform ratios do not significantly change in response to HFD feeding, in contrast to males. Accordingly, the mRNA expression levels of the adipogenic transcription factors are maintained upon HFD feeding in wt females. Whether this sex-specific difference might be due to the less pronounced inflammation (macrophage infiltration) observed in females or to other sex-specific responses to HFD feeding has to be examined in future studies. Probably, the slight increase in PPARγ expression together with the increased LAP function might be sufficient for the observed adipocyte hyperplasia in female *Cebpb^ΔuORF* mice on HFD, which however, seems to be less pronounced compared to HFD fed *Cebpb^ΔuORF* males. The downregulation of fatty acid synthetase (FAS) mRNA levels that we observe in wt mice upon HFD seems to be independent from the expression of the adipogenic transcription factors tested because at least in females these regulatory events were uncoupled and might be due to other effects of HFD feeding. Furthermore, in *Cebpb^ΔuORF* females the protein expression of FAS on HFD does not correspond to the FAS mRNA levels, it is efficiently reduced despite almost completely maintained mRNA levels suggesting interfering, post-transcriptional effects. What these effects are and why they are only observed in female mice is so far not known.

Taken together, our data propose pharmacological reduction of LIP expression as an approach to switch the unhealthy metabolic phenotype of obese individuals into a healthy obese phenotype to prevent the development of metabolic disease (possibly together with pharmacologic PPARγ activation). We have shown that a search for such an intervention is feasible through the identification of drugs that inhibit LIP expression similar to mTORC1-inhibition (*Zaini et al., 2017*). One drug that we identified as an inhibitor of LIP expression, the antiviral drug adevovir dipivoxil (*Zaini et al., 2017*), was recently tested on female wt mice upon ND and HFD feeding (*Bitto et al., 2021*). This study showed that adevovir is effective in increasing the LAP/LIP isoform ratio also in vivo (although in the mice LIP expression was not affected but rather LAP expression was increased), which resulted in increased C/EBPβ target gene activation and increased expression of β-oxidation genes in the liver similar to what we observed in the *Cebpb^ΔuORF* mice (*Zidek et al., 2015*). Remarkably, adevovir treatment resulted in a significant reduction of body weight and fat content particularly in the HFD fed mice (*Bitto et al., 2021*) similar to what we found with the *Cebpb^ΔuORF* females on HFD. Whether this effect on body weight and fat accumulation is caused only by increasing C/EBPβ function or whether additional not yet identified effects of adevovir contribute in addition is not known so far.

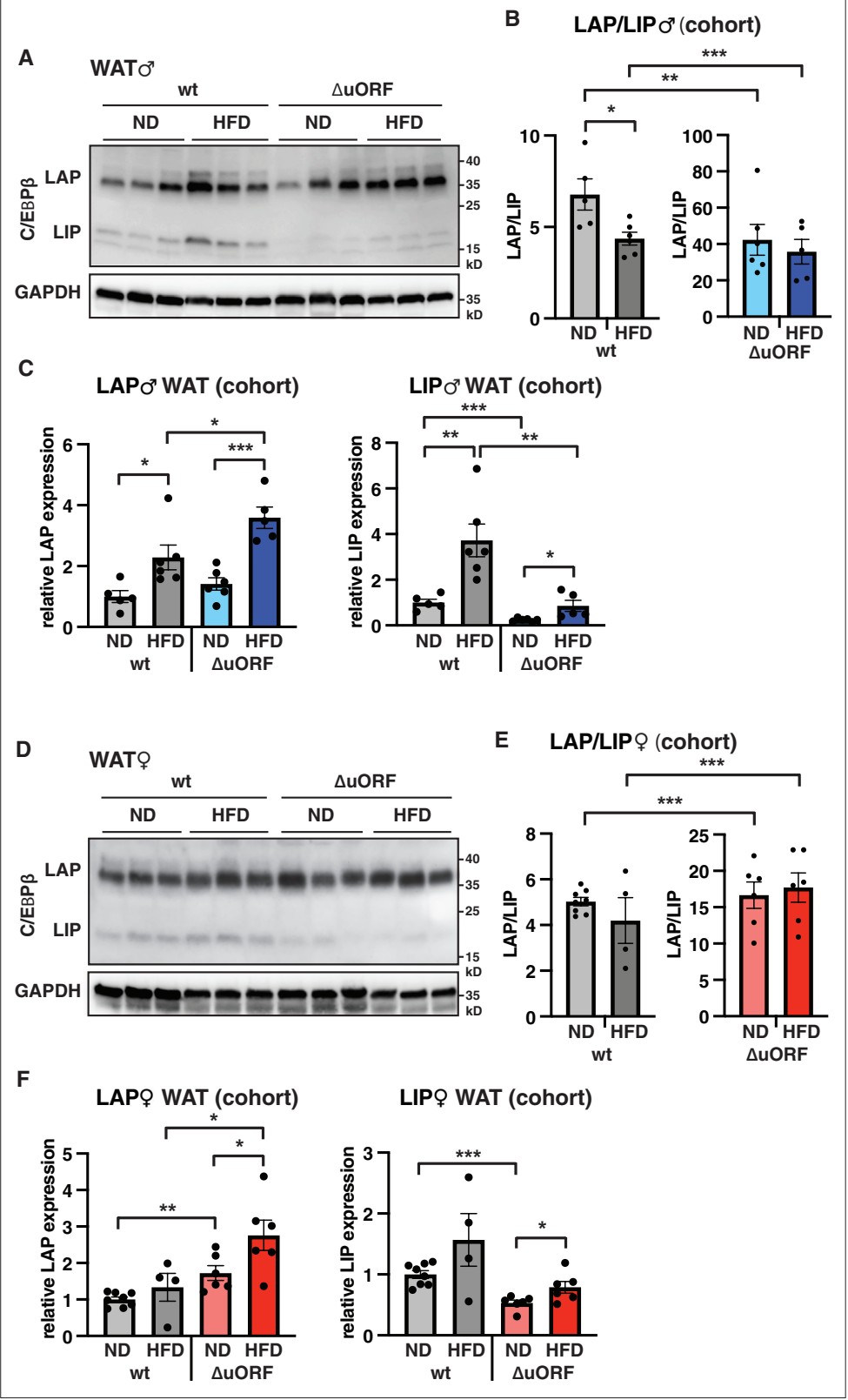

**Figure 8.** LAP and LIP expression under ND and HFD feeding. (**A**) Immunoblots of C/EBPβ and GAPDH loading control performed with visceral WAT extracts from wt or *Cebpb*$^{\Delta uORF}$ males on either normal diet (ND) or HFD (19 weeks). (**B**) Quantification of the LAP/LIP ratio in split bar diagrams for better visualization, and (**C**) quantification of LAP and LIP isoform expression separately (normalized to the GAPDH signal) using the whole

*Figure 8 continued on next page*

*Figure 8 continued*

cohort (wt ND, n = 5; wt HFD, n = 6; ΔuORF ND, n = 6; ΔuORF HFD, n = 5). (**D**) Immunoblots of C/EBPβ and GAPDH loading control performed with visceral WAT extracts from wt or *Cebpb^ΔuORF^* females on either normal diet (ND) or HFD (19 weeks). (**E**) Quantification of the LAP/LIP ratios in split bar diagrams for better visualization, and (**F**) quantification of LAP and LIP isoform expression separately (normalized to the GAPDH signal) using the whole cohort (wt ND, n = 8; wt HFD, n = 4; ΔuORF ND and HFD, n = 6). All values are mean ± SEM. p-Values were determined with Student's t-test, *p < 0.05; **p < 0.01; ***p < 0.001.

The online version of this article includes the following source data for figure 8:

**Source data 1.** Raw data related to *Figure 8B, C, E and F*.

Furthermore, it will be interesting to examine whether adevovir treatment of males results in reduced fat accumulation like in females or in increased (subcutaneous) fat accumulation as we observed in the *Cebpb^ΔuORF^* males. Since the increased C/EBPβ function in *Cebpb^ΔuORF^* mice also has the potential to extend the healthspan (*Müller et al., 2018*), therapeutic interference with the C/EBPβ isoform ratio may represent a promising strategy to attenuate the metabolic disturbances not only associated with overnutrition but also with ageing.

# Materials and methods

## Key resources table

| Reagent type (species) or resource | Designation | Source or reference | Identifiers | Additional information |
|---|---|---|---|---|
| Genetic reagent (*M. musculus*) | C/EBPβ^ΔuORF^ | https://doi.org/10.1101/gad.557910 https://doi.org/10.15252/embr.201439837 | | males, back-crossed for 6 generations and females, back-crossed for 12 generations into C57BL/6 J background |
| Antibody | Anti-C/EBPβ (E299) (rabbit monoclonal) | Abcam | Cat# ab32358, RRID:AB_726796 | (1:1000) |
| Antibody | Anti-C/EBPα (D56F10) (rabbit monoclonal) | Cell Signaling | Cat# 8178, RRID:AB_11178517 | (1:1000) |
| Antibody | Anti-PPARγ (C26H12) (rabbit monoclonal) | Cell Signaling | Cat# 2435, RRID:AB_2166051 | (1:1000) |
| Antibody | Anti-FAS (C20G5) (rabbit monoclonal) | Cell Signaling | Cat# 3180, RRID:AB_2100796 | (1:1000) |
| Antibody | Anti-GAPDH (14 C10) (rabbit monoclonal) | Cell Signaling | Cat# 2118, RRID:AB_561053 | (1:1000) |
| Antibody | Anti-SREBP1 (2 A4) (mouse monoclonal) | NeoMarkers | Cat# MS-1207-PO | (1:1000) |
| Antibody | Anti-CD68 (E307V) (rabbit monoclonal) | Cell Signaling | Cat# 97,778 | (1:200) |
| Antibody | Anti-rabbit IgG, HRP-conjugated (donkey polyclonal) | GE Healthcare | Cat#: NA934, RRID:AB_772206 | (1:5000) |
| Antibody | Anti-mouse IgG, HRP-conjugated (sheep polyclonal) | GE Healthcare | Cat#: NXA931, RRID:AB_772209 | (1:5000) |
| Antibody | Anti-rabbit IgG, biotin-conjugated (goat polyclonal) | Vector Labs | Cat#: BA-1000 | (1:250) |
| Sequence-based reagent | CD68 (F) | https://doi.org/10.7554/eLife.34985.001 | PCR primer | 5'-GCCCACCAC CACCAGTCACG –3' |
| Sequence-based reagent | CD68 (R) | https://doi.org/10.7554/eLife.34985.001 | PCR primer | 5'GTGGTCCAG GGTGAGGGCC A-3' |
| Sequence-based reagent | PPARγ (F) | https://doi.org/10.15252/embr.201439837 | PCR primer | 5'-GCCCTTTGG TGACTTTATGG –3' |
| Sequence-based reagent | PPARγ (R) | https://doi.org/10.15252/embr.201439837 | PCR primer | 5'-CAGCAGGTT GTCTTGGATGT 3' |
| Sequence-based reagent | C/EBPα (F) | https://doi.org/10.15252/embr.201439837 | PCR primer | 5'-CAAGAACAG CAACGAGTACC G-3' |
| Sequence-based reagent | C/EBPα (R) | https://doi.org/10.15252/embr.201439837 | PCR primer | 5'-GTCACTGGT CAACTCCAGCA C-3' |

*Continued on next page*

*Continued*

| Reagent type (species) or resource | Designation | Source or reference | Identifiers | Additional information |
|---|---|---|---|---|
| Sequence-based reagent | SREBP1c (F) | https://doi.org.10.15252/embr.201439837 | PCR primer | 5'-AACGTCACT TCCAGCTAGAC –3' |
| Sequence-based reagent | SREBP1c (R) | https://doi.org.10.15252/embr.201439837 | PCR primer | 5'-CCACTAAGG TGCCTACAGAG C-3' |
| Sequence-based reagent | FAS (F) | https://doi.org.10.15252/embr.201439837 | PCR primer | 5'-ACACAGCAA GGTGCTGGAG-3' |
| Sequence-based reagent | FAS (R) | https://doi.org.10.15252/embr.201439837 | PCR primer | 5'-GTCCAGGCT GTGGTGACTCT –3' |
| Sequence-based reagent | TNFα (F) | This paper | PCR primer | 5'-CCAGACCCT CACACTCA-3' |
| Sequence-based reagent | TNFα (R) | This paper | PCR primer | 5'-CACTTGGTG GTTTGCTACGA C-3' |
| Sequence-based reagent | MCP1 (F) | This paper | PCR primer | 5'-GCTGGGAGAG CTACAAGAGGA TCA-3' |
| Sequence-based reagent | MCP1 (R) | This paper | PCR primer | 5'-ACAGACCTC TCTCTTGAGCT TGGT-3' |
| Sequence-based reagent | IL-1β (F) | This paper | PCR primer | 5'-GAAATGCCA CCTTTTGACAG TG-3' |
| Sequence-based reagent | IL-1β (R) | This paper | PCR primer | 5'-TGGATGCTC TCATCAGGACA G-3' |
| Sequence-based reagent | IL-6 (F) | This paper | PCR primer | 5'-CCGGAGAGG AGACTTCACAG –3' |
| Sequence-based reagent | IL-6 (R) | This paper | PCR primer | 5'-TTCTGCAAG TGCATCATCGT –3' |
| Sequence-based reagent | GAPDH (F) | This paper | PCR primer | 5'-ATTGTCAGC AATGCATCCTG –3' |
| Sequence-based reagent | GAPDH (R) | This paper | PCR primer | 5'-ATGGACTGT GGTCATGAGC C-3' |
| Sequence-based reagent | β-actin (F) | https://doi.org.10.15252/embr.201439837 | PCR primer | 5'-AGAGGGAAA TCGTGCGTGA C-3' |
| Sequence-based reagent | β-actin (R) | https://doi.org.10.15252/embr.201439837 | PCR primer | 5'-CAATAGTGA TGACCTGGCC GT-3' |
| Commercial assay or kit | Vectastain ABC HRP Kit | Vector Labs | Cat#: PK-4000 | |
| Commercial assay or kit | Western Lightning Plus ECL Reagent | Perkin Emer | Cat#: NEL103001EA | |
| Commercial assay or kit | ECL Prime Western Blotting Reagent | GE Healthcare | Cat#: RPN2236 | |
| Commercial assay or kit | Restore Western Blot Stripping buffer | Thermo Fisher | Cat#: 21,063 | |
| Commercial assay or kit | QIAzol Lysis re-agent | QIAGEN | Cat#: ID:79,306 | |
| Commercial assay or kit | RNeasy Lipid Tissue Mini kit | QIAGEN | Cat#: ID:74,804 | |
| Commercial assay or kit | Rneasy Plus Mini kit | QIAGEN | Cat#: ID:74,134 | |
| Commercial assay or kit | Transcriptor First Strand cDNA Synthesis kit | Roche | Cat#: 4379012001 | |
| Commercial assay or kit | Light Cycler 480 SYBR Green I Master Mix | Roche | Cat#: 0470751600 | |
| Chemical compound, drug | Insulin (human) | Lilly | Cat#: HI-210 | |
| Chemical compound, drug | Sudan III | Sigma-Aldrich | Cat#: S4136 | |
| Chemical compound, drug | Oil-Red-O | Sigma-Aldrich | Cat#: O0625 | |
| Software, algorithm | GraphPad Prism 9.0 | Graphpad Software, La Jolla, CA | RRID:SCR_002798 | |
| Software, algorithm | Image Quant LAS 4000 Mini Imager Software | GE Healthcare | RRID:SCR_014246 | |
| Software, algorithm | ImageJ | https://doi.org.10.1186/s12859-017-1934-z | RRID:SCR_003070 | |

## Mice

*Cebpb^ΔuORF* mice (**Wethmar et al., 2010**) were back-crossed for 6 generations (males) or for 12 generations (females) into the C57BL/6 J background. Mice were kept at a standard 12 hr light/dark cycle at 22 °C in a pathogen-free animal facility and for all experiments age-matched mice were used. Mice were fed a high-fat diet (HFD; 60% fat, D12492, Research Diets New Brunswick, USA) for 19 weeks starting at an age of 12–15 weeks or a standard chow diet (normal diet, ND; 10% fat, D12450B, Research Diets New Brunswick, USA) as control. For each genotype, weight-matched mice were

distributed over the different diet groups. Mice were analyzed at different time points as indicated in the figure legends. The determination of male body weight and food intake (per cage divided through the number of mice in the cage) was performed weekly for 16 or 18 weeks, respectively. The body weight of females was determined in week 19 after mice were terminated. During the performance of all experiments the genotype of the mice was masked. All animal experiments were performed in compliance with protocols approved by the Institutional Animal Care and Use committee (IACUC) of the Thüringer Landesamt für Verbraucherschutz (#03-005/13).

## Determination of body composition

Mice were anesthetized and the abdominal region from lumbar vertebrae 5–6 was analyzed using an Aloka LaTheta Laboratory Computed Tomograph LCT-100A (Zinsser Analytic) as described in *Zidek et al., 2015*.

## Determination of caloric utilization

Both the feces and samples of the HFD food were collected, dried in a speed vacuum dryer at 60 °C for 5 hr, grinded and pressed into tablets. The energy content of both the feces and food samples was determined through bomb calorimetry using an IKA-Calorimeter C5000. The energy efficiency was calculated through subtraction of the energy loss in the feces from the energy consumed.

## IP glucose tolerance and insulin sensitivity tests

For the determination of glucose tolerance, mice were starved overnight (16 hr) and a 20% (w/v) glucose solution was injected i.p., using 10 µl per gram body weight. After different time points, the blood glucose concentration was measured using a glucometer (AccuCheck Aviva, Roche). For the determination of insulin sensitivity, an insulin solution (0,05 IU/ml insulin in 1xPBS supplemented with 0.08% fatty acid-free BSA) was i.p. injected into non-starved mice using 10 µl per gram body weight and the blood glucose concentration was measured as described above.

## Histological staining

Tissue pieces were fixed for 24 hr with paraformaldehyde (4%) and embedded in paraffin. Sections (5 µm) were stained with hematoxylin and eosin (H&E) in the Autostainer XL (Leica). Adipocyte area was determined using the ImageJ software from 12 adjacent cells per mouse. For CD68 staining, sections (5 µm) from paraffin embedded tissue were dried for 2 hr at 55 °C, deparaffinized and rehydrated. For antigen retrieval, sections were incubated for 25 min in 10 mM citrate buffer, pH 6.0. Endogenous peroxidase was blocked in 1% $H_2O_2$ in methanol for 30 min. After blocking with normal goat serum (1:10 in PBS), sections were incubated with a CD68 specific antibody (E307V, #97,778 from Cell Signaling, 1:200) over night at 4 °C followed by incubation with a biotin-conjugated secondary goat anti rabbit antibody (Vector Labs, BA-1000, 1:250) for 30 min and incubation with reagents of the Vectastain ABC HRP kit (Vector Labs, PK-4000) according to the manufacturer's instruction. Slides were stained with DAB and counterstained with hematoxylin, dehydrated and covered using Eukitt. A Hamamatsu scanner was used to take images. For lipid staining with Sudan III, cryosections (10 µm) were fixed with paraformaldehyde (4%) and stained with Sudan-III solution (3% (w/v) Sudan-III in 10% ethanol and 90% acetic acid) for 30 min. For lipid staining with Oil-Red-O, cryosections (10 µm) were fixed with paraformaldehyde (10%), washed shortly in 60% isopropanol and stained with Oil-Red-O solution (3 mg/ml isopropanol stock solution diluted to 1.8 mg/ml with $H_2O$) for 15 min. After shortly washing first with isopropanol and then with water, cells were counterstained with hematoxylin and covered with 10 mM Tris HCl pH 9.0 in glycerol.

## Calculation of adipocyte number

The mean adipocyte area from the visceral fat per mouse was used to calculate the adipocyte volume per mouse with r (radius) = and V (volume) = $\pi r^3$. For males, the mean volume of the visceral fat as determined by CT analysis was then divided by the mean adipocyte volume to get the cell number. For females, adipocyte weight was calculated by multiplying the calculated cell volume with 0.915 (the density of triolein). Then, the mean weight of the visceral WAT tissue was divided by the calculated adipocyte weight.

## Determination of organ weight

After termination of the mice organs were collected and cleaned from surrounding fat or connective tissue and their weight was determined using an analytical balance.

## qRT-PCR analysis

Tissue pieces were homogenized using the Precellys 24 system (Peqlab) in the presence of 1 ml QIAzol reagent (QUIAGEN). The RNA was isolated using the RNeasy Lipid Tissue Mini kit (QUIAGEN) according to the protocol of the manufacturer, incubated with RQ1 RNase-free DNase (Promega) for 30 min at 37 °C and purified further using the RNeasy Plus Mini kit (QUIAGEN) starting from step 4.

One μg RNA was reverse transcribed into cDNA with Oligo(d)T primers using the Transcriptor First Strand cDNA Synthesis kit (Roche). The qRT-PCR was performed with the LightCycler 480 SYBR Green I Master mix (Roche) using the following primer pairs: CD68: 5'-GCC CAC CAC CAC CAG TCA CG-3' and 5'-GTG GTC CAG GGT GAG GGC CA-3', PPARγ 5'-GCC CTT TGG TGA CTT TAT GG-3' and 5'-CAG CAG GTT GTC TTG GAT GT-3', C/EBPα 5'-CAA GAA CAG CAA CGA GTA CCG-3' and 5'-G TC ACT GGT CAA CTC CAG CAC-3', SREBP1c: 5'-AAC GTC ACT TCC AGC TAG AC-3' and 5'-CCA CTA AGG TGC CTA CAG AGC-3', FAS: 5'-ACA CAG CAA GGT GCT GGA G-3' and 5'-GTC CAG G CT GTG GTG ACT CT-3', TNFα: 5'-CCA GAC CCT CAC ACT CA-3' and 5'-CAC TTG GTG GTT TGC TAC GAC-3', MCP1: 5'-GCT GGA GAG CTA CAA GAG GAT CA-3' and 5'-ACA GAC CTC TCT CTT GAG CTT GGT, IL-1β: 5'-GAA ATG CCA CCT TTT GAC AGT G-3' and 5'-TGG ATG CTC TCA TCA G GA CAG-3', IL-6: 5'-CCG GAG AGG AGA CTT CAC AG-3' and 5'-TTC TGC AAG TGC ATC ATC GT-3', GAPDH: .5'- ATTGTCAGCAATGCATCCTG-3' and 5'- ATGGACTGTGGTCATGAGCC-3' and β-actin: 5'-AGA GGG AAA TCG TGC GTG AC-3' and 5'-CAA TAG TGA TGA CCT GGC CGT-3'.

## Immunoblot analysis

Tissues were lysed in RIPA buffer as described in *Müller et al., 2018*. Equal amounts of protein were separated by SDS-PAGE and transferred to a PDVF membrane. For protein detection, the following antibodies were used: C/EBPβ (E299, ab32358, 1:1000) from Abcam, C/EBPα (D56F10, #8178, 1:1000), PPARγ (C26H12, #2435, 1:1000), FAS (C20G5, #3180, 1:1000) and GAPDH (14C10, #2118, 1:1000) from Cell Signaling, SREBP1 2A4, MS-1207-PO, 1:1000 from NeoMarkers, and HRP-linked anti rabbit or mouse IgG from GE Healthcare. For detection, Lightning Plus ECL reagent (Perkin Elmer) or ECL prime reagent (GE Healthcare) was used. For re-probing, the membranes were incubated for 15 min with Restore Western Blot Stripping buffer (Thermo Fisher). The Image Quant LAS Mini 400 Imager or the Image Quant 800 Imager (both GE Healthcare) were used for detection and quantification of C/EBPβ LAP and LIP isoforms was performed using the supplied software.

## Statistical methods

The number of biological replicates is indicated as n = x. All graphs show average ± standard error of the mean (SEM). To calculate statistical significance of the obtained results the Student's t-Test was used with *p < 0.05; **p < 0.01 and ***p < 0.001. Single mice were excluded when results indicated technical failure of the experimental performance. Furthermore, extreme outliers were excluded from the analysis.

## Acknowledgements

We thank Susanne Klaus and Susanne Keipert (DIfE, Potsdam) for help with bomb calorimetry and Maaike Oosterveer (UMCG) for providing the SREBP1 antibody. At the FLI, Verena Kliche for technical assistance, the staff of the animal house facility for embryo transfer and advice on mouse experiments, and Maik Baldauf for help with histology. L.M.Z. was supported by the Deutsche Forschungsgemeinschaft (DFG) through a grant (CA 283/1–1) to C.F.C.

## Additional information

### Funding

| Funder | Grant reference number | Author |
|---|---|---|
| Deutsche Forschungsgemeinschaft | CA 283/1-1 | Laura M Zidek |

The funders had no role in study design, data collection and interpretation, or the decision to submit the work for publication.

### Author contributions

Christine Müller, Laura M Zidek, Conceptualization, Data curation, Formal analysis, Investigation, Methodology, Project administration, Supervision, Validation, Visualization, Writing - original draft, Writing - review and editing; Sabrina Eichwald, Gertrud Kortman, Mirjam H Koster, Formal analysis, Investigation, Methodology; Cornelis F Calkhoven, Conceptualization, Formal analysis, Funding acquisition, Methodology, Project administration, Supervision, Validation, Visualization, Writing - original draft, Writing - review and editing

### Author ORCIDs

Christine Müller (iD) http://orcid.org/0000-0003-1974-4053
Cornelis F Calkhoven (iD) http://orcid.org/0000-0001-6318-7210

### Ethics

All animal experiments were performed in compliance with protocols approved by the Institutional Animal Care and Use committee (IACUC) of the Thüringer Landesamt für Verbraucherschutz (#03-005/13).

### Decision letter and Author response

Decision letter https://doi.org/10.7554/eLife.62625.sa1
Author response https://doi.org/10.7554/eLife.62625.sa2

## Additional files

### Supplementary files

• Transparent reporting form

### Data availability

Source data are included in the Source Data files.

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
