## [Editor Report]

This study provides important insight into the mechanisms involved in regulating the response to an obesity-inducing diet in mice. This study demonstrates that C/EBPβ acts as a key protective factor against many of the negative consequences of a high-fat diet in mice and further clarifies the downstream processes involved. Obesity is an already significant and growing health problem, and this work may help identify new strategies to combat obesity going forward.

---

## [Decision Letter]

**Decision letter after peer review:**

Thank you for submitting your article "C/EBPβ regulates hyperplastic versus hypertrophic fat tissue growth" for consideration by *eLife*. Your article has been reviewed by 2 peer reviewers, and the evaluation has been overseen by a Reviewing Editor and Matt Kaeberlein as the Senior Editor. The reviewers have opted to remain anonymous.

The reviewers have discussed the reviews with one another and the Reviewing Editor has drafted this decision to help you prepare a revised submission.

Summary:

There is consensus among the reviewers that this study provides an interesting and important advance in understanding the role of CEBP/b LAP in metabolism and response to a high fat diet challenge. The major discoveries reported here are fairly well supported by the data, including that male uORF KO mice show an increase in fat cell number as opposed to fat cell size, less inflammation, and improved glucose tolerance/insulin sensitivity.

Essential revisions:

However, all of the reviewers shared the major concern that it appears that only male mice were studied here, and that this fact – or the rationale for using only male mice – was not clearly articulated within the manuscript. This makes interpretation quite challenging, especially given that the authors previously published that lifespan extension in the uORF KO mice is much more pronounced in female compared to male mice. There was consensus that this is a substantial weakness to the current manuscript which limits its overall impact. It's possible the authors already have this data, and we would need to see inclusion of data supporting similar outcomes for the key experiments in female mice to recommend publication in *eLife*. If the outcomes are different in males and females, this is likely quite interesting and would need to be developed further.

The other significant concern was related to the RT-qPCR data, which is indicative but not conclusive support for the authors' conclusions, especially since many of the changes are small in magnitude. It was noted that most of the relevant proteins have ELISAs available, and they all have antibodies which could be used to support the robustness and importance of the small but plausibly important differences observed. IHC against CD68 in the fat depots could be performed and the authors could strengthen their claims about adipose tissue inflammation by measuring the expression levels of inflammatory cytokines in adipose depots.

---

## [Author Response]

Essential revisions:However, all of the reviewers shared the major concern that it appears that only male mice were studied here, and that this fact – or the rationale for using only male mice – was not clearly articulated within the manuscript. This makes interpretation quite challenging, especially given that the authors previously published that lifespan extension in the uORF KO mice is much more pronounced in female compared to male mice. There was consensus that this is a substantial weakness to the current manuscript which limits its overall impact. It's possible the authors already have this data, and we would need to see inclusion of data supporting similar outcomes for the key experiments in female mice to recommend publication in eLife. If the outcomes are different in males and females, this is likely quite interesting and would need to be developed further.The other significant concern was related to the RT-qPCR data, which is indicative but not conclusive support for the authors' conclusions, especially since many of the changes are small in magnitude. It was noted that most of the relevant proteins have ELISAs available, and they all have antibodies which could be used to support the robustness and importance of the small but plausibly important differences observed.IHC against CD68 in the fat depots could be performed and the authors could strengthen their claims about adipose tissue inflammation by measuring the expression levels of inflammatory cytokines in adipose depots.

We now included data of female mice complementary to most of the originally performed experiments for males.

Please find all changes and additions:

Title: we propose a more extended version of the title.

Bar graphs: in the various figures we now grouped genotypes instead of diet type for – in our opinion – easier assessment.

Figure 1: panels E-G now show new data from female mice.

The body weight of female mice was obtained at the end of the experiment upon termination of the mice (Figure 1E). Due to our move to a different institute, we could not perform body composition analysis of females by micro-CT as we did with males and therefore used the weights of visceral and subcutaneous fat obtained from isolated fat tissue from terminated mice at the end of the experiment (Figure 1F, G).

Figure supplement 1: no changes.

We could not include results of food intake and energy efficiency from female mice.

Figure 2: panel A shows male data from previous Figure 1E and panel B show new data from females.

Figure 3: Panel A was shown in previous Figure 1F. The panels B and C shows new data for males.

IHC against CD68 in the visceral fat depot were performed that strengthen the claim about macrophage infiltration in the visceral adipose tissue (B). In addition, we included qPCR analyses of the inflammatory cytokines TNFα, MCP1, IL-1β an IL-6 in visceral fat from males (C).

Figure 4: Shows new data from females complementary to the male data in Figure 3.

Also here, qPCR analysis of CD68 expression (C), IHC against CD68 (B) and qPCR analyses of the inflammatory cytokines TNFα, MCP1, IL-1β an IL-6 in the visceral fat depot were performed.

Figure 5: Panel A and C show male data previously shown in Figure 2A and figure 2 supplement 1A. Panels B and D show new data for females.

The lipid staining in livers from female was performed using Oil-Red-O (Figure 5B) instead of Sudan III that was used for males.

Figure 5 supplement 1: Panels A, B and C show male data previously shown in Figure 2 B, C and Figure 2 supplement 1 B. Panel D shows new data from females.

Figure 6: Panel A and C show data from males previously shown in Figure 3. Panels B and D show new data from females.

Figure 7: Panel A shows data from males previously shown in Figure 4. Panel B shows data from females.

Figure 7 supplement 1: shows new data for males (A) and females (B).

We show immunoblot analysis of genes whose expression was different between the diets as determined by qPCR analysis.

Figure 8: shows all new data for males and females. We show immunoblots of C/EBPβ isoform expression in extracts from visceral fat of males (A) and females (D) and quantifications of the LAP/LIP isoform ratio (B, E) and the LAP and LIP isoforms separately (C, F).

Discussion: The section has been extended based on added results and suggestions by the reviewers and parts of the discussion on the connection of our findings to the ageing process was removed to limit the extend of this section and to focus more on HFD feeding and obesity.

Material and methods: experimental details have been supplemented based on added data.